# Using machine learning for detection of Parkinson's disease and mild cognitive impairment

Anthaea-Grace Patricia Dennis[1,2]*, Sarah L. Martin[3,4], Robert Chen[1,2,5],
Philip Gerretsen[2,3,6], Antonio P. Strafella[1,2,3,5]*

1 Krembil Brain Institute, University Health Network, University of Toronto, Ontario, Canada, 2 Institute of Medical Science, Temerty Faculty of Medicine, University of Toronto, Ontario, Canada, 3 Brain Health Imaging Centre, Campbell Family Mental Health Research Institute, Centre for Addiction and Mental Health, University of Toronto, Ontario, Canada, 4 Translational and Computational Neuroscience Unit, Faculty of Health and Education, Manchester Metropolitan University, Manchester, United Kingdom, 5 Edmond J. Safra Parkinson Disease Program & Morton and Gloria Shulman Movement Disorder Unit, Neurology Division, Dept. of Medicine, Toronto Western Hospital, University Health Network, University of Toronto, Ontario, Canada, 6 Department of Psychiatry, University of Toronto, Ontario, Canada

* agp.dennis@mail.utoronto.ca (AGPD); antonio.strafella@utoronto.ca (APS)

## Abstract

### Background

Parkinson's disease is a movement disorder featuring motor symptoms and cognitive decline, which can manifest as mild cognitive impairment. The incidence of mild cognitive impairment increases with disease progression, and Parkinson's disease can cause significant disability, therefore, identification of Parkinson's disease and mild cognitive impairment in Parkinson's disease is imperative. Neuroimaging and biofluid biomarkers have been studied separately, however, research suggests that combining biomarkers may improve detection.

### Objectives

We aimed to investigate using machine learning whether a combination of neuroimaging and biofluid biomarkers would result in more effective identification of Parkinson's disease and mild cognitive impairment.

### Methods

Utilizing the Parkinson's Progression Markers Initiative dataset, we applied two different machine learning approaches, support vector machine and random forest, to explore combinations of neuroimaging and cerebrospinal fluid biomarkers for detection.

**Data availability statement:** Code for this study is publicly available from the Zenodo repository (https://doi.org/10.5281/zeno-do.17195713). Data for this study cannot be shared by the authors because of PPMI's Data Usage Agreements, which prevent users to re-publish data. All data used in this study, as well as a data dictionary, are free and publicly available at the PPMI website (www.ppmi-info.org/access-data-specimens/download-data), upon an online application and the signature of the Data User Agreement and of the publications policies. For any questions on the PPMI Intellectual Property (IP) Policy or applying for an exception to the PPMI IP Policy, researchers may contact them via email (ppmi@michaelj-fox.org). The authors declare that they did not have any special access privileges that others would not have when attempting to access the data from PPMI.

**Funding:** A-G.P.D. is supported by the Queen Elizabeth II Graduate Scholarship in Science & Technology (https://www.sgs.utoronto.ca/awards/queen-elizabeth-ii-graduate-scholar-ship-in-science-technology/), the Krembil Brain Institute (https://www.uhn.ca/Krembil), and the University of Toronto School of Graduate Studies (https://www.sgs.utoronto.ca/). A.P.S. is supported by the Canadian Institute of Health Research (CIHR) (PJT-173540) (https://cihr-irsc.gc.ca/e/193.html) and the Krembil-Rossy Chair program (https://rossy-foundation.org/; https://www.uhn.ca/Krembil). These funders did not play any role in the study design, data collection and analysis, decision to publish, or preparation of the manuscript.

**Competing interests:** The authors have declared that no competing interests exist.

## Results

Overall, both machine learning techniques had an equivalent performance. In general, in those models for detecting Parkinson's disease, DaT-SPECT performed better than biofluid biomarkers. In models for detecting Parkinson's disease patients with mild cognitive impairment, combining DaT-SPECT with phosphorylated-tau-181 resulted in higher accuracy, outperforming DaT-SPECT alone.

## Conclusions

Classification for Parkinson's disease and mild cognitive impairment may be improved by combining neuroimaging with biofluid biomarkers through machine learning models.

---

## Introduction

Parkinson's disease (PD) is the second-most common neurodegenerative disorder and affects approximately 2% of adults over the age of 65 globally [1]. PD is primarily characterized by motor symptoms such as bradykinesia, resting tremor, and muscular rigidity. However, patients with PD may develop a decline in cognitive abilities as well [2].

Over time, cognitive decline can result in mild cognitive impairment (MCI), and eventually dementia (within PD, dementia is referred to as PD dementia). The likelihood of developing MCI increases with disease duration, disease severity, and age, affecting approximately 19–62% of individuals with PD [3–4]. The incidence of progression from MCI to PD dementia is suggested to be approximately 25–62%, depending on when follow-up was performed [3]. Therefore, identifying biomarkers for PD and MCI is essential for improving diagnosis and for predicting the onset and progression of cognitive decline.

Dopamine transporter single photon emission computed tomography (DaT-SPECT) is commonly used as a biomarker and diagnostic tool for parkinsonism [5]. DaT-SPECT uses the $^{123}$I-ioflupane radiopharmaceutical, which binds reversibly to dopamine transporters in the striatum [5]. This technique allows for the identification of hallmark degeneration within the dopaminergic system in the PD brain, thereby supporting diagnosis. While DaT-SPECT is highly sensitive to PD pathology, research suggests that combining it with other imaging modalities, such as MRI or CT, can improve diagnostic accuracy [6]. However, using multiple neuroimaging modalities may be impractical for some patients, and can yield inconsistent results due to differences in patient population and methodologies [7]. In addition, the pathology of cognitive impairment is associated with other neurotransmitters and brain regions, especially deficits in the cholinergic system [8], thus DaT-SPECT is generally not suitable for the identification of MCI in people with PD. Therefore, detection of PD-MCI or PD dementia likely requires the combination of several biomarkers.

Cerebrospinal fluid (CSF) biomarkers for neurodegenerative disease and cognitive decline are promising tools, as approximately 20% of CSF proteins are derived from cells of the central nervous system [9]. CSF-based biomarkers relevant to parkinsonism and cognitive decline include alpha-synuclein (α-syn), beta-amyloid-42 (Aβ42), total-tau (t-tau), phosphorylated-tau (p-tau), and neurofilament light (NfL) [9]. Specifically, lower levels of α-syn for PD, lower levels of Aβ42 for cognitive decline, and elevated levels of p-tau, t-tau, and NfL for cognitive decline have been reported consistently [10–11]. In PD and dementia with Lewy bodies, misfolded α-syn causes toxicity and aggregates in neurons [12]. The α-syn propagation has additionally been found to be possibly correlated with development of cognitive impairment [10]. A 2015 study on α-syn in CSF found that while no significant differences were found between PD and dementia with Lewy bodies, the α-syn level was significantly lower in PD as compared to Alzheimer's subjects [13]. Thus, it is possible that a similar difference may be detected between PD and other forms of cognitive impairment, and that the use of CSF α-syn levels may improve classification of PD patients with cognitive impairment or dementia.

Additionally, coexistent β-amyloid and tau pathologies are commonly found in neurodegenerative disorders. α-syn can interact with proteins such as beta-amyloid or tau, and these proteins can influence each other in their pathologies [14]. Studies have found that β-amyloid and tau (which are both involved in Alzheimer's pathology) are correlated to severity of cognitive impairment in PD dementia [14]. Additionally, studies have observed that Aβ42 levels were lower in PD-MCI or PD dementia than in PD with normal cognition [10], and that p-tau levels were elevated in PD dementia or dementia with Lewy bodies compared to patients with PD and normal cognition [15]. NfL can differentiate PD from atypical parkinsonisms [15]. In response to axonal damage, high levels of NfL are released into the CSF and the blood, although the concentration of NfL in the blood is 40-fold lower than in the CSF [16]. Higher plasma and CSF NfL levels are potentially useful diagnostic tools for PD, and prognostic biomarker for PD-MCI conversion to PD dementia [11]. Based on this, NfL combined with DaT-SPECT may improve diagnostic accuracy of PD and MCI.

Artificial intelligence and machine learning techniques have been used to detect and test different biomarkers [17–18]. These tools can identify differences in biomarker level between patient groups (e.g., healthy controls and PD) which may aid with diagnosis and track disease progression. Here, we applied support vector machine (SVM) and random forest (RF) machine learning techniques to investigate whether the combination of DaT-SPECT imaging with CSF biomarkers may improve the detection of PD and MCI. We predicted that the combination of DaT-SPECT and CSF biomarkers would improve the model performance and thus help distinguish PD with normal cognition (NC) from PD with MCI and healthy controls (HC).

## Methodology

### Participant and data collection

Data used in the preparation of this article were obtained from the Parkinson's Progression Markers Initiative (PPMI) database (www.ppmi-info.org/data) on November 22, 2023 [19]. For up-to-date information on the study, visit www.ppmi-info.org.

All the participating PPMI sites received approval from an ethical standards committee on human experimentation before study commencement, received informed written consent from all participants in the study, and was in full compliance with the principles set out by the Declaration of Helsinki. Authors did not have access to information that could identify individual participants during or after data collection. Subjects considered for this study met inclusion and exclusion criteria identified by the PPMI protocol available online at https://www.ppmi-info.org/sites/default/files/docs/PPMI002_Clinical%20Protocol_AM3.2_30Jan2023_Final.pdf.

In addition to the inclusion and exclusion criteria set by the PPMI protocol, participants were excluded if they were found to have a genetic marker associated with PD risk, including GBA, LRRK2, PARKIN, PINK1, or SNCA, since these markers may result in a different disease progression and may add an additional confounding variable to the machine learning models [20]. This excluded $n = 2077$ subjects. Additionally, as subjects with REM sleep disorder (RBD) may have

a different disease trajectory and have different symptoms as compared to subjects without, we excluded these subjects ($n = 283$). The remaining cohort of subjects included $n = 537$ subjects.

The inclusion criteria for this study were: (1) at least one instance of DaT-SPECT imaging data for the left and right caudate and the left and right putamen (excluded $n = 37$ subjects), (2) at least two instances of data for one of five cerebrospinal fluid-based biomarkers, including α-syn, Aβ42, t-tau, p-tau, and NfL (excluded $n = 211$ subjects), and (3) two Montreal Cognitive Assessment (MoCA) scores collected at different time points, with one score assigned as a baseline score (excluded $n = 12$ subjects). Subjects were excluded if they had MCI at baseline (MoCA score of ≤ 25) (excluded $n = 13$ subjects). From these MoCA scores, PD and HC subjects were classified as converting to MCI if the second score collected was ≤ 25. This created two groups; 1) those who did not convert to MCI and retained normal cognition (NC), and 2) those who did convert to MCI. The DaT-SPECT imaging was collected in the first visit, and the second MoCA score and secondary CSF biomarker data were collected in the second visit. The average amount of time from first timepoint to the second timepoint was 45.74 months ± 22.39 months. Finally, subjects were excluded if they had missing data ($n = 10$).

After applying these criteria, the final sample included 254 subjects: 41 PD subjects with NC, 62 PD subjects with MCI, 83 HC subjects with NC, and 68 HC subjects with MCI. We defined healthy controls as subjects who did not have PD, and this cohort included both subjects with and without MCI. We did not exclude HC with MCI, since cognitive change is a normal part of aging and including this ensures that any models created using the full HC sample can be generalized and applied in broader settings.

## Clinical assessments

Demographic information assessed included age, sex, year of education, race, family history of PD, and handedness. Clinical assessments included the Movement Disorders Society-Unified Parkinson Disease Rating Scale (MDS-UPDRS) part III motor section [21] "ON" medication, Hoehn and Yahr score [22], and disease duration. Global cognition was assessed with the MoCA [23]. The MoCA is a widely available and quick to administer test with good sensitivity for detecting cognitive impairment in PD [23].

## Statistical analyses

We analysed demographic and clinical data to assess significant differences based on PD or MCI presence. We assumed that the variance around the mean for each group was similar, data were obtained randomly, were independent, and were normally distributed. We used the Pearson's Chi-squared test with Yates' continuity correction for sex, as it was a categorical variable. Fisher's exact test examined differences in race, family history of PD, and handedness. Continuous variables of demographic, clinical measures, and biomarker data were assessed via a two-tailed t-test. Additionally, for multiple groups, we used ANOVA and the Tukey HSD test for post-hoc analysis of the continuous variables. In all tests, an alpha value of 0.05 was used as the p-value threshold for significance.

## Biomarkers and machine learning techniques

As biomarkers of interest, we considered DaT-SPECT imaging and biofluid data (CSF: α-syn, Aβ42, t-tau, p-tau, NfL) acquired at approved PPMI centers using standardized protocols (available online at https://www.ppmi-info.org/sites/default/files/docs/PPMI2.0_SPECT_TOM_Final_v6.0_20221201_FE.pdf and https://www.ppmi-info.org/sites/default/files/docs/PPMI%20Biologics%20Manual%20of%20Procedures%20V12.pdf, respectively).

To determine the efficacy of the proposed biomarkers in detecting PD and MCI, we used two different machine learning techniques: support vector machine (SVM) and random forest (RF).

SVM is a commonly used technique that determines the hyperplane (a line in a high-dimensional space) best able to separate the data into classifications, in this case, PD and HC, or MCI and NC [17,24]. The hyperplane is selected so that the distance between the data of both classes and the hyperplane is maximized [17,24]. Based on SVM's mechanism,

this technique is used for binary classification. To classify between *n* classes, (*n*-1)! different SVMs must be trained. For instance, classification between 4 classes would require 6 SVMs [17,24]. By comparison, RF is based on the decision tree structure, which uses the value of an input variable to divide the original dataset into smaller sets, which are then further separated by other variables until the sets consist entirely of one class of subjects (e.g., PD) [17]. The basic decision tree (DT) technique can easily cause overfitting with many variables, so the RF technique has been developed as an improvement. In RF, multiple DTs are created, with each using a different randomly chosen input variable. Majority vote is used for the final classification [25]. Since SVM and RF are two different techniques with different mechanisms, we compared the performance of models using SVM and RF [17]. The model training and testing process was conducted through the R programming language's (version 4.3.2) caret package [26–27], which was created specifically for machine learning.

All models included the demographics listed in Table 1. We first trained and tested models for detecting PD and MCI using each of the biomarkers alone. These sets of models will be referred to as "*singular*" models, because they each use a singular biomarker (DaT, α-syn, Aβ42, t-tau, p-tau or NfL). We compared performance for each of the biomarkers and determine which were most effective. We used 10-fold cross-validation and a training/test split of 70%/ 30% to avoid overfitting and erroneous model performance [17]. The model training process is additionally visualized in Fig 1.

In the models used for detecting PD, the DaT-SPECT biomarker included SBRs for the caudate and the entire putamen. In the models used for detecting MCI, the biomarker included striatal binding ratios (SBRs) for the caudate and anterior putamen [28]. In all models, the CSF biomarkers included the biomarker level at the baseline, the biomarker level at the secondary timepoint, and the rate of change of the biomarker level per month, calculated from the baseline and secondary biomarker levels (Table 2).

We created singular models for three different purposes: to identify PD in a set of PD vs HC subjects; to identify MCI in a set of HC subjects (HC-NC vs HC-MCI); and to identify MCI in a set of PD subjects (PD-NC vs PD-MCI). For the first classification, we wanted to identify which biomarkers were useful for identification of PD regardless of cognitive class, so that a model could be applied to any patient in a clinical setting to diagnose PD. For the second and third classifications, we separated the sample by presence of PD, since the development of MCI may be different in HC compared to PD.

Since DaT-SPECT may not reliably distinguish PD-NC from PD-MCI and may not identify all subjects [29] and using multiple neuroimaging modalities may not be feasible [7,30], we investigated whether a CSF biomarker could assist in classification and identify subjects that DaT-SPECT did not. We predicted that the combination of DaT-SPECT and CSF biomarkers (*combined models*) would improve the model performance closer to 100%, and thus identify MCI in PD, which would distinguish PD-NC from PD-MCI or PDD, and HC. To assess this, we trained new models on 70% of all data each with one of the two or more biomarkers we aimed to combine. The predictions of these models were used to train new "combined" models on 15–20% of data (termed the validation set). These new combined models were then tested on the remaining 10–15% of data, which included nine or ten subjects due to the smaller cohort of subjects with multiple CSF biomarker data and used DaT-SPECT with one or two of the CSF biomarkers. The combinations we tested for these

**Table 1. Demographics included in machine learning models.**

| # | Metric | Variable type |
|---|--------|---------------|
| 1 | Age | Continuous |
| 2 | Years of Education | Discrete |
| 3 | Sex (Male or Female) | Nominal – Binary |
| 4 | Handedness (Left, Right, or Mixed) | Nominal |
| 5 | Race (White, Black, Mixed, or Asian) | Nominal |
| 6 | Family History of Parkinson's Disease (Yes or No) | Nominal – Binary |
| 7 | Body Mass Index | Continuous |

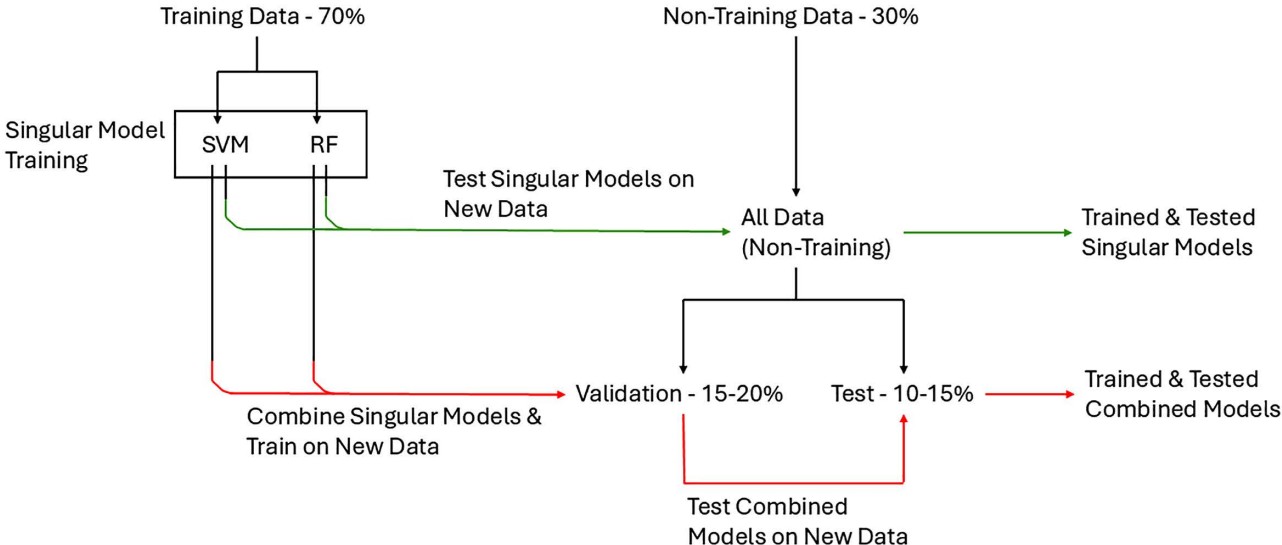

**Fig 1. Workflow for model training and testing process.** For both SVM and RF models classifying based on a singular biomarker (green pipeline), they are trained on 70% of data and tested on all 30%. In comparison, for models combining two or more biomarkers, models are first created from one biomarker and trained on this data (70%). Then, these models are used to train a new model with multiple biomarkers with the validation set of data (15-20%). The combined models are then tested on the remaining set of data (10-15%). *Abbreviations: SVM – support vector machine; RF – random forest.*

**Table 2. Biomarkers included in machine learning models.**

| # | Metric (Sample Size Available) | Variable Type | Description | Sample Size | | | |
|---|---|---|---|---|---|---|---|
| | | | | PD-NC | PD-MCI | HC-NC | HC-MCI |
| 1 | DaT-SPECT (*n* = 253) | Continuous | SBRs in the left and right caudate and the left and right putamen OR anterior putamen* | 41 | 62 | 82 | 68 |
| 2 | CSF – alpha-synuclein (*n* = 248) | Continuous | Baseline and secondary CSF conc. and rate of change of alpha-synuclein (pg/mL) | 40 | 62 | 80 | 66 |
| 3 | CSF – beta-amyloid-42 (*n* = 240) | Continuous | Baseline and secondary CSF conc. and rate of change of beta-amyloid-42 (pg/mL) | 38 | 62 | 77 | 63 |
| 4 | CSF – total-tau (*n* = 250) | Continuous | Baseline and secondary CSF conc. and rate of change of total-tau (pg/mL) | 41 | 62 | 82 | 65 |
| 5 | CSF –phosphorylated-tau-181 (*n* = 237) | Continuous | Baseline and secondary CSF conc. and rate of change of phosphorylated-tau-181 (pg/mL) | 36 | 60 | 78 | 63 |
| 6 | CSF – neurofilament light (*n* = 157) | Continuous | Baseline and secondary CSF conc. and rate of change of neuro-filament light (pg/mL) | 27 | 38 | 48 | 44 |

Abbreviations: DaT-SPECT – dopamine transporter SPECT; SBRs – striatal binding ratios; CSF – cerebrospinal fluid. Conc. – concentration. * When detecting PD, the SBRs for entire putamen were used; when detecting MCI, SBRs for the anterior putamen were used.

models are described in Table 3. The model combinations reported in the article had at least one metric above the 80% threshold (explained below).

When combining biomarkers, we tested models for four different purposes: to identify PD in a cohort of NC subjects (HC-NC vs PD-NC); to identify PD in a cohort of MCI subjects (HC-MCI vs PD-MCI); to identify MCI in a set of HC subjects (HC-NC vs HC-MCI); and to identify MCI in a set of PD subjects (PD-NC vs PD-MCI). The first two classifications were aimed at identifying PD in either subjects with NC or subjects with MCI. We separated the sample into subjects with NC and subjects with MCI instead of using the larger PD vs HC classification so that we could identify the influence of cognitive impairment on the

**Table 3. Biomarker combinations tested in machine learning models.**

| # | CSF marker(s) used | |
|---|---|---|
| 1 | α-syn | * |
| 2 | NfL | |
| 3 | Aβ-42 | * |
| 4 | p-tau | * |
| 5 | t-tau | * |
| 6 | α-syn + NfL | * |
| 7 | Aβ-42 + α-syn | |
| 8 | Aβ-42 + t-tau | * |
| 9 | Aβ-42 + p-tau | * |
| 10 | Aβ-42 + NfL | |
| 11 | t-tau + α-syn | |
| 12 | t-tau + NfL | |
| 13 | t-tau + p-tau | |
| 14 | p-tau and NfL | |

Abbreviations: DaT-SPECT – dopamine transporter SPECT; CSF – cerebrospinal fluid, α-syn; Alpha-synuclein; Aβ42; Beta-amyloid-42; t-tau; total-tau; p-tau; phosphorylated-tau-181; NfL; neurofilament light. All combinations used DaT-SPECT.

* Well performing model combinations.

detection of PD, and the best way to do this was to minimize any confounders and control for changes in cognition. Additionally, it was possible that different CSF biomarkers would be relevant in NC subjects as compared to MCI subjects. Finally, the last two classifications (HC-NC vs HC-MCI and PD-NC vs PD-MCI) were the same as in the singular models.

## Machine learning metrics

The five metrics collected to measure model performance were accuracy, area under the receiver operating characteristic curve (AUC), Cohen's kappa, sensitivity, and specificity. Sensitivity is defined as the "ability of a test to correctly classify an individual as diseased,", calculated as (# of true positives)/ (# of true positives + # of false negatives) [31]. In comparison, specificity is defined as "the ability of a test to correctly classify an individual as disease-free," calculated as (# of true negatives)/ (# of true negatives + # of false positives) [31]. The AUC uses randomized thresholds for the variable level required to classify subjects as positive to determine sensitivity and specificity. For example, a threshold of 0 used for a variable ranging from 0 to 1, where subjects above the threshold are classified as positive, would result in sensitivity of 100% and specificity of 0%. Inversely, a threshold of 1 would result in sensitivity of 0% and specificity of 100%. After determining the sensitivity and specificity for the thresholds, the area under the curve created by plotting sensitivity and (1-specificity) can be used to measure the model performance [17]. Kappa measures the agreement of the model's prediction with the actual classifications [32]. If the agreement is perfect, kappa is 1, if predictions and reality are in total disagreement, kappa is −1, and if prediction is random, kappa is 0. To have a uniform metric to determine whether a model performed well or not, we used a threshold of 80%, where if all metrics were above the 80% threshold, the model performed well [33]. For simplicity, we will report mainly the findings above the 80% threshold.

## Results

### Demographics and clinical and biomarker characteristics

A general summary of demographic and clinical characteristics between PD and HC groups are provided in Table 4a. The main differences in demographic and clinical characteristics among PD-NC, PD-MCI, HC-NC, and HC-MCI groups are highlighted in Table 4b and Table 4c. Additionally, we investigated the DaT-SPECT and CSF biomarker profiles for

statistical differences across groups. S1a and S1b Tables provide a summary of the main differences between PD and HC and between the PD-NC, PD-MCI, HC-NC, and HC-MCI cohorts.

## Models using singular biomarkers

**Models intended for PD detection.** In models aimed to differentiate PD from HC regardless of cognitive status, DaT-SPECT demonstrated the most accurate performance compared to the CSF biomarkers (Table 5). Both SVM and RF models using DaT-SPECT performed exceptionally well with accuracy values close to 100%. DaT-SPECT had high sensitivity (94.33% in SVM and 92.33% in RF) and high specificity (96.89% in SVM and 96.00% in RF), meaning that the majority of PD subjects and HC were correctly identified (Table 5, highlighted in bold).

In contrast, the CSF biomarkers were poor at identifying subjects with PD due to low sensitivity, yet had good metrics for specificity, indicating that the models performed well when identifying HC. For example, NfL had poor sensitivity (34.21% in SVM and 48.95% in RF), but had higher specificity (91.11% in SVM and 74.81% in RF).

The additional statistical analysis of the biomarkers between groups (S1a Table) supported these observations, with DaT-SPECT showing significant differences between PD and HC, whereas CSF biomarkers and their rates of change (particularly for α-syn) were not consistently different between PD and HC. Overall, the SVM and RF modelling approaches provided similar results with minor differences in outcome measures, and as expected, DaT-SPECT was the best biomarker for detecting PD (Table 5).

**Models intended for MCI detection.** SVM and RF models aimed for detecting MCI (HC-NC vs HC-MCI; PD-NC vs PD-MCI) did not perform well, with performance below 80% (S2 and S3 Tables). Regardless of which machine learning technique used, performance was consistently poor. These observations were not entirely surprising given the lack of significant difference for DaT-SPECT and for the CSF biomarkers' rates of change between those cohorts (S1b Table).

## Models using biomarker combinations

**Models intended for PD detection in subjects with NC.** In SVM and RF models differentiating subjects with PD-NC vs HC-NC, DaT-SPECT alone remained the best choice for classification (Table 6). In Table 6, we reported the performance of the model detecting PD-NC using DaT-SPECT alone only for comparison. The addition of α-syn and NfL biomarkers did not improve the models' ability to classify subjects as PD-NC and HC-NC, as the accuracy using DaT-SPECT was 95.83%, and the accuracy using DaT+α-syn or DaT+α-syn+NfL were both lower (80% in SVM and 90% in RF; 62.5% in SVM and 87.5% in RF, respectively). Overall, these observations were consistent with the statistical analyses of the biomarkers between those cohorts, with DaT-SPECT showing a significant difference between PD-NC vs HC-NC, whereas CSF biomarkers and rate of change were not consistently different between groups (S1b Table). In general, RF performed better than SVM, regardless of the biomarker combination used (Fig 2, Table 6).

**Models intended for PD detection in subjects with MCI.** In SVM and RF models contrasting PD-MCI vs HC-MCI, the performance obtained using DaT-SPECT alone was improved by the addition of CSF-biomarkers (Table 7, Fig 3). In particular, models created with the combinations of DaT+Aβ42, DaT+p-tau and DaT+Aβ42+p-tau performed similarly at 100% in all accuracy metrics, and these models outperformed the performance of DaT-SPECT alone (Table 7, see highlighted in bold). Due to the high performance, we also investigated the confidence intervals for these 100% results, which are reported in the table. The biomarker statistical analysis showed that only DaT-SPECT and CSF p-tau were significantly different between groups (HC-MCI vs PD-MCI) (S1b Table), thus suggesting this to be the better combination option. For all the other biomarker combinations, SVM performed better than RF (Fig 3, Table 7).

**Models intended for MCI detection.** When detecting MCI (HC-NC vs HC-MCI and PD-NC vs PD-MCI), SVM and RF models did not perform well with a performance below 80%. Accuracy values ranged from 0–70% and 40–75% in HC and PD, respectively (S4 and S5 Tables). These observations were not entirely surprising given that DaT-SPECT and CSF

**Table 4.** a. Population demographics and clinical features. b. Continuous demographics and clinical variables of cohort subgroups. c. Categorical demographics of cohort subgroups .

**a. Population demographics and clinical features.**

| Metric | PD (n=103) | HC (n=151) | p-value | df | Test Statistic |
|---|---|---|---|---|---|
| Population (Male/Female) | 63/40 | 98/53 | 0.635 | 1 | χ=0.225 |
| Age | 63.8 (9.5) | 61.4 (10.9) | 0.068 | 236.68 | t=−1.8323 |
| Handedness (Right/Mixed/Left) | 93/2/8 | 121/10/20 | 0.074 | N/A | N/A |
| Education (Years) | 15.7 (2.7) | 16.2 (3.0) | 0.173 | 230.05 | t=1.366 |
| Race (White/Black/Mixed/Asian) | 94/3/4/2 | 139/7/4/1 | 0.699 | N/A | N/A |
| Family History of PD (Yes/No) | 24/79 | 9/142 | **<.001** | N/A | N/A |
| Disease Duration (Years) | 0.44 (0.46) | N/A | N/A | N/A | N/A |
| MDS UPDRS-III | 23.0 (10.3) | 1.7 (2.6) | **<.001** | 111.02 | t=−20.528 |
| Hoehn and Yahr Score | 1.9 (0.5) | 0.1 (0.3) | **<.001** | 158.35 | t=−33.324 |
| MoCA – Baseline Observation | 28.0 (1.3) | 28.2 (1.1) | 0.176 | 188.02 | t=1.357 |
| MoCA – Second Observation | 24.8 (4.6) | 26.5 (2.8) | **0.001** | 155.62 | t=3.280 |

**b. Continuous demographics and clinical variables of cohort subgroups**

| Variable | | Groups | | | ANOVA (df=3*) | | | Post-Hoc P-value | |
|---|---|---|---|---|---|---|---|---|---|
| | PD-NC (n=41) | PD-MCI (n=62) | HC-NC (n=83) | HC-MCI (n=68) | | PD-NC vs PD-MCI | HC-NC vs HC-MCI | HC-NC vs PD-NC | HC-MCI vs PD-MCI |
| Age | 60.88 (10.63) | 65.67 (8.24) | 58.14 (11.21) | 65.39 (8.98) | **F=9.84 p<.001** | 0.078 | **<.001** | 0.465 | 0.999 |
| Education (Yrs) | 15.71 (2.82) | 15.73 (2.70) | 16.14 (3.19) | 16.15 (3.06) | F=0.63 p=0.594 | – | – | – | – |
| Disease Duration (Yrs) | 0.42 (0.45) | 0.47 (0.48) | N/A | N/A | F=3.44 p=0.066 | – | N/A | N/A | N/A |
| MDS-UPDRS-III | 19.9 (9.8) | 25.2 (10.1) | 1.4 (2.3) | 2.1 (3.0) | **F=212.86 p<.001** | **<.001** | 0.924 | **<.001** | **<.001** |
| H&Y | 1.8 (0.4) | 1.9 (0.5) | 0.1 (0.3) | 0.1 (0.3) | **F=441.69 p<.001** | 0.102 | 1.00 | **<.001** | **<.001** |
| Baseline MoCA | 28.4 (1.3) | 27.8 (1.3) | 28.4 (1.2) | 28.0 (0.9) | **F=4.74 p=0.003** | – | – | – | – |
| Second MoCA | 28.6 (1.3) | 22.3 (4.2) | 28.6 (1.5) | 23.8 (1.6) | **F=110.4 p<.001** | **<.001** | **<.001** | 0.999 | **0.003** |

**c. Categorical demographics of cohort subgroups**

| Variable | | Groups | | | Chi-Squared/ Fisher's Test (df=3) | | | Post-Hoc Testing (df=1) | |
|---|---|---|---|---|---|---|---|---|---|
| | PD-NC n=41 | PD-MCI n=62 | HC-NC n=83 | HC-MCI n=68 | | PD-NC vs PD-MCI | HC-NC vs HC-MCI | HC-NC vs PD-NC | HC-MCI vs PD-MCI |
| Population (Male/Female) | 18/23 | 45/17 | 48/35 | 50/18 | **χ=17.833 p<.001** | **χ=7.380 p=0.007** | χ=3.384 p=0.066 | χ=1.616 p=0.204 | χ=0 p=1 |
| Handedness (Right/Mixed/Left)* | 37/0/4 | 56/2/4 | 72/5/6 | 49/5/14 | **p=0.049** | p=0.538 | **p=0.042** | p=0.290 | **p=0.033** |

*(Continued)*

**Table 4.** (Continued)

| Race (White/Black/Mixed/Asian)* | 37/3/1/0 | 57/0/3/2 | 78/1/3/1 | 61/6/1/0 | p = 0.074 | – | – | – | – |
|---|---|---|---|---|---|---|---|---|---|
| Family History of PD (Yes/No)* | 14/27 | 10/52 | 8/75 | 1/67 | **p = 0.021** | p = 0.055 | **p = 0.042** | **p = 0.002** | **p = 0.003** |

Abbreviations: PD; Parkinson's Disease, HC; Healthy controls, df – Degrees of Freedom; N/A – not applicable. MoCA; Montreal Cognitive Assessment, MDS-UPDRS; Movement Disorder Society-Unified Parkinson's Disease Rating Scale part III. Statistical significance (p < .05) highlighted in bold. Continuous data presented as mean (standard deviation).

Abbreviations: PD – Parkinson's Disease, NC – normal cognition, df – degrees of freedom, MCI – Mild cognitive impairment, HC – Healthy controls. H&Y – Hoehn & Yahr Score. Statistical significance (p < .05) highlighted in bold. All variables are reported with Mean (Standard Deviation). * Disease Duration has df of 1.

Abbreviations: PD; Parkinson's Disease, NC; normal cognition, MCI; Mild cognitive impairment, HC; Healthy controls. Statistical significance (p < .05) highlighted in bold. * These analyses used Fisher's test, which does not generate a test statistic.

**Table 5.** Singular Models – Metric performance for SVM and RF in PD vs HC.

| Model | ML | AUC | ACC | KPA | SNS | SPC |
|---|---|---|---|---|---|---|
| DaT | **SVM** | **0.99** | **95.87%** | **0.91** | **94.33%** | **96.89%** |
|  | **RF** | **0.99** | **94.53%** | **0.89** | **92.33%** | **96.00%** |
| NfL | SVM | 0.70 | 67.61% | 0.28 | 34.21% | 91.11% |
|  | RF | 0.67 | 64.13% | 0.24 | 48.95% | 74.81% |
| Aβ42 | SVM | 0.69 | 62.78% | 0.16 | 25.33% | 89.52% |
|  | RF | 0.65 | 62.92% | 0.22 | 45.67% | 75.24% |
| α-syn | SVM | 0.66 | 63.97% | 0.18 | 26.67% | 90.00% |
|  | RF | 0.68 | 65.34% | 0.27 | 51.67% | 74.89% |
| p-tau | SVM | 0.61 | 65.57% | 0.19 | 22.86% | 94.05% |
|  | RF | 0.63 | 63.29% | 0.21 | 45.00% | 75.48% |
| t-tau | SVM | 0.62 | 62.30% | 0.14 | 25.00% | 87.73% |
|  | RF | 0.61 | 61.22% | 0.18 | 44.33% | 73.73% |

Abbreviations: ML – machine learning technique; SVM – support vector machine; RF – random forest; PD – Parkinson's disease; DaT – dopamine transporter-SPECT imaging; α-syn – alpha-synuclein; Aβ42 – beta-amyloid-42; t-tau – total-tau; p-tau – phosphorylated-tau-181; NfL – neurofilament light; ACC – accuracy, AUC – area under the curve; KPA – kappa; SNS – sensitivity; SPC – specificity.

Best performing biomarker highlighted in bold.

biomarkers' rates of change were not statistically different between cohorts (S1b Table). Thus, whether using SVM or RF models, none of the combined biomarkers improved detection of MCI.

## Discussion

In this study, we attempted to determine the feasibility of using SVM and RF machine learning models to test the efficacy of the different biomarkers (i.e., DaT-SPECT imaging and CSF: α-syn, Aβ42, t-tau, p-tau, NfL) for detecting PD and MCI. We tested these biomarkers both individually and in different combinations. Overall, the two main observations were that 1) SVM and RF often had an equivalent performance and 2) model performance was excellent when biomarkers were statistically different among groups. The latter point influenced which biomarker combination led to better model performance.

In more detail, we created models first to identify PD in a set of PD vs HC subjects (regardless of cognitive status) testing biomarkers individually. Models using DaT-SPECT performed exceptionally well using either SVM or RF with high accuracy (Table 5, highlighted in bold). DaT-SPECT had high sensitivity (94.33% in SVM and 92.33% in RF) and high specificity (96.89% in SVM and 96.00% in RF), suggesting that the majority of PD subjects and HC were correctly

**Table 6. Combined Models – Metric performance for SVM and RF in PD-NC vs HC-NC.**

| Model | ML | AUC | ACC | KPA | SNS | SPC |
|---|---|---|---|---|---|---|
| DaT | **SVM** | **0.99** | **95.83%** | **0.90** | **90.00%** | **98.75%** |
| | **RF** | **0.99** | **95.83%** | **0.91** | **91.67%** | **97.92%** |
| DaT & α-syn | SVM | 0.79 | 80.00% | 0.58 | 83.33% | 75.00% |
| | RF | 0.88 | 90.00% | 0.78 | 100.00% | 75.00% |
| DaT, α-syn, & NfL | SVM | 0.70 | 62.50% | 0.33 | 100.00% | 40.00% |
| | RF | 0.93 | 87.50% | 0.75 | 100.00% | 80.00% |

Abbreviations: ML – machine learning technique; SVM – support vector machine; RF – random forest; PD – Parkinson's disease; NC – normal cognition; DaT – dopamine transporter-SPECT imaging; α-syn – alpha-synuclein; NfL – neurofilament light; ACC – accuracy, AUC – area under the curve; KPA – kappa; SNS – sensitivity; SPC – specificity.

Best performing biomarker highlighted in bold.

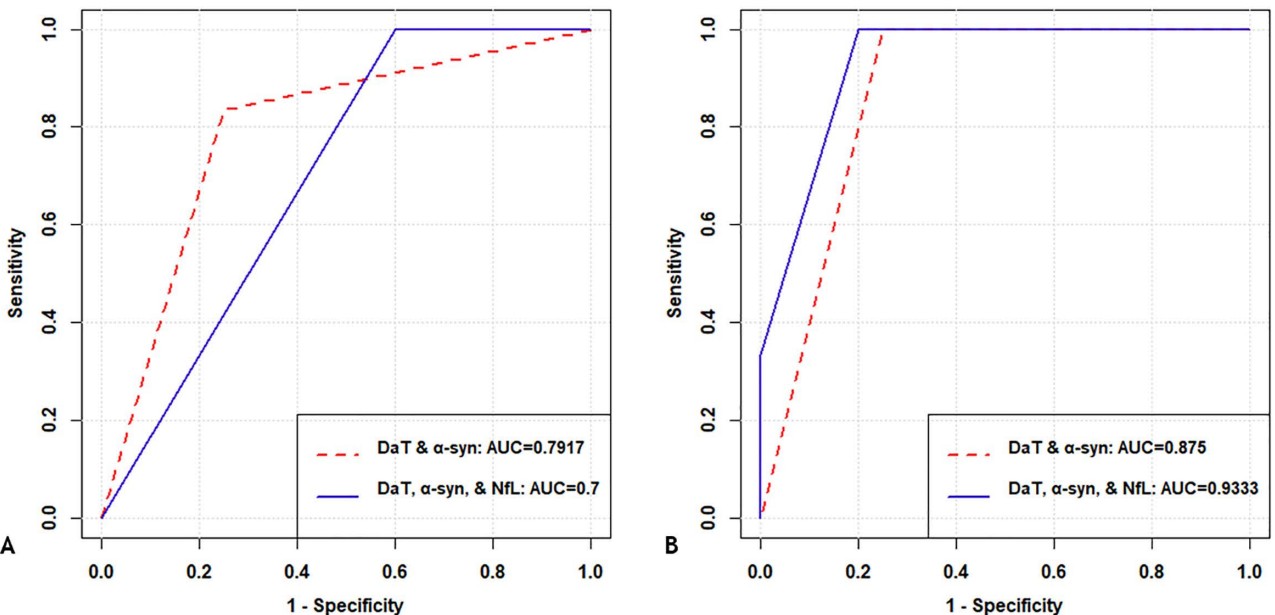

**Fig 2. Combined models – Model performance in PD-NC vs HC-NC. (A) Using SVM. (B) Using RF.** All metrics are reported in Table 6. Abbreviations: ROC: receiver operating characteristic curve; SVM: support vector machine; RF: random forest; PD: Parkinson's Disease; NC: normal cognition; DaT: dopamine transporter-SPECT imaging; α-syn: alpha-synuclein; NfL: neurofilament light; AUC: area under the curve; ACC: accuracy; KPA: kappa; SNS: sensitivity; SPC: specificity.

identified. In contrast, the CSF biomarkers performed poorly with both SVM and RF. Overall, these observations were not surprising given that DaT-SPECT, which measures presynaptic dopaminergic degeneration, showed significant differences between PD and HC, whereas CSF biomarkers and their rate of change (particularly for α-syn) were not consistently different between PD and HC (S1a Table).

We also tested individual biomarkers to detect MCI both in HC (HC-NC vs HC-MCI) and PD cohorts (PD-NC vs PD-MCI). All models performed poorly, below the 80% threshold, regardless of which machine learning technique was used (S2 and S3 Tables). In general, these observations were consistent with the lack of significance for DaT-SPECT and CSF biomarker's rate of change between cohorts (S1b Table).

**Table 7. Combined models – Metric performance for SVM and RF in PD-MCI vs HC-MCI.**

| Model | ML | AUC | ACC | 95% CI | KPA | SNS | SPC |
|---|---|---|---|---|---|---|---|
| DaT | SVM | 0.98 | 92.37% | – | 0.85 | 92.78% | 92.00% |
| | RF | 0.97 | 89.47% | – | 0.79 | 87.78% | 91.00% |
| DaT & Aβ42 | **SVM** | **1** | **100%** | **(0.6915, 1)** | **1** | **100%** | **100%** |
| | **RF** | **1** | **100%** | **(0.6915, 1)** | **1** | **100%** | **100%** |
| DaT, Aβ42, & p-tau | **SVM** | **1** | **100%** | **(0.7684, 1)** | **1** | **100%** | **100%** |
| | **RF** | **1** | **100%** | **(0.7684, 1)** | **1** | **100%** | **100%** |
| DaT & p-tau | **SVM** | **1** | **100%** | **(0.6915, 1)** | **1** | **100%** | **100%** |
| | **RF** | **1** | **100%** | **(0.6915, 1)** | **1** | **100%** | **100%** |
| DaT, Aβ42, & t-tau | SVM | 0.86 | 85.71% | – | 0.71 | 71.43% | 100.00% |
| | RF | 0.84 | 71.43% | – | 0.43 | 42.86% | 100.00% |
| DaT & t-tau | SVM | 0.8 | 80% | – | 0.6 | 100.00% | 60.00% |
| | RF | 0.74 | 80% | – | 0.6 | 100.00% | 60.00% |

Abbreviations: ML – machine learning technique; SVM – support vector machine; RF – random forest; PD – Parkinson's disease; DaT – dopamine transporter-SPECT imaging; Aβ42 – beta-amyloid-42; t-tau – total-tau; p-tau – phosphorylated-tau-181; ACC – accuracy, AUC – area under the curve; KPA – kappa; SNS – sensitivity; SPC – specificity. Best performing biomarker highlighted in bold.

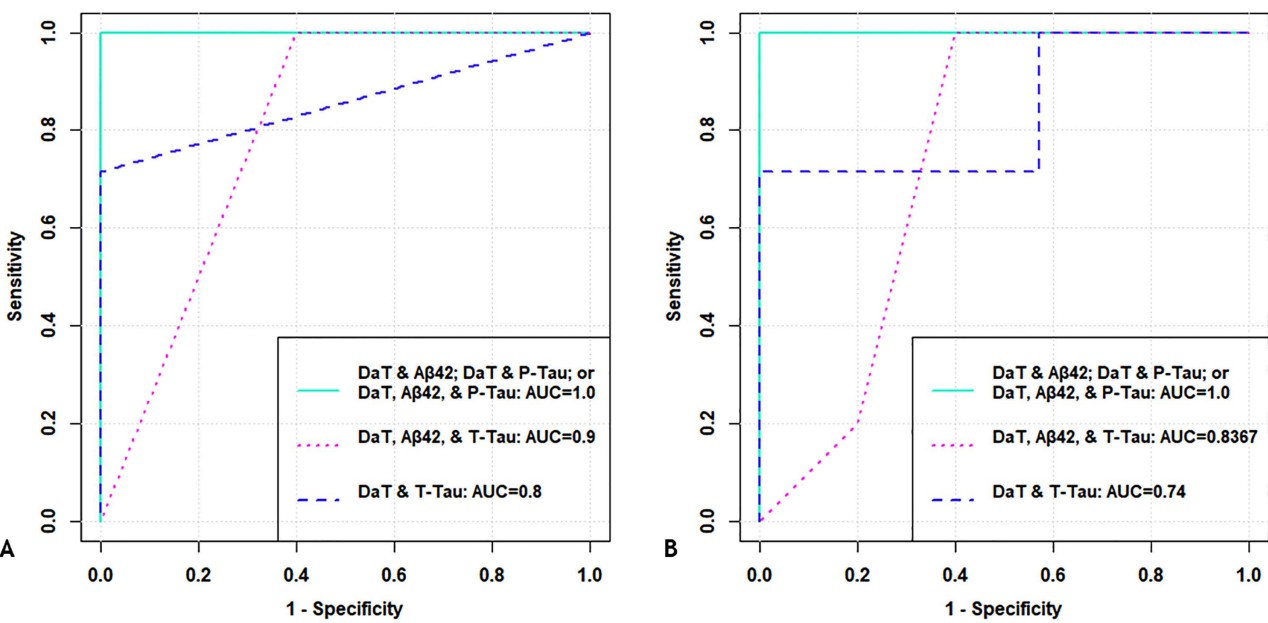

**Fig 3. Combined Models – Model performance in PD-MCI vs HC-MCI. (A) Using SVM. (B) Using RF.** All metrics are reported in **Table 7**. Abbreviations: ROC: receiver operating characteristic curve; SVM: support vector machine; RF: random forest; PD: Parkinson's Disease; MCI: mild cognitive impairment; DaT: dopamine transporter-SPECT imaging; Aβ42: beta-amyloid-42; P-Tau: phosphorylated-Tau; T-Tau: total-Tau; AUC: area under the curve; ACC: accuracy; KPA: kappa; SNS: sensitivity; SPC: specificity.

As noted above, different combinations of biomarkers were also tested to identify PD in a cohort of NC subjects (HC-NC vs PD-NC) and in a cohort of MCI subjects (HC-MCI vs PD-MCI). We separated the sample into subjects with NC and subjects with MCI so that we could control for the influence of cognitive impairment on the detection of PD. In these analyses, for PD detection in subjects with NC (HC-NC vs PD-NC), the addition of α-syn and NfL biomarkers (to DaT-SPECT)

did not significantly improve the SVM or RF models' ability to detect PD. DaT-SPECT alone still remained the best choice for classification with an accuracy of 95.83% for both SVM and RF (Table 6, highlighted in bold). As noted above, these observations were consistent with the significant differences in DaT-SPECT between PD-NC vs HC-NC, while CSF biomarkers and their rates of change were not consistently different between groups (S1b Table). Overall, RF performed better than SVM, regardless of the biomarker combination used (Fig 2, Table 6). In models intended for PD detection in subjects with MCI (HC-MCI vs PD-MCI), the DaT-SPECT performance was improved by the addition of CSF-biomarkers (Fig 3, Table 7). In particular, models created with the combinations of DaT+Aβ42, DaT+p-tau and DaT+Aβ42+p-tau performed similarly at 100% in all accuracy metrics, and these models well outperformed the performance of DaT-SPECT alone (Table 7, see highlighted in bold). Consistent with these models' observations, DaT-SPECT and CSF p-tau bio-markers were significantly different between groups (HC-MCI vs PD-MCI), thus suggesting that the DaT+p-tau and DaT+Aβ42+p-tau combinations are better options (S1b Table). For the best performing biomarker combinations, RF and SVM performed equally well (Fig 3, Table 7).

In identifying MCI in a set of HC subjects (HC-NC vs HC-MCI) and in a set of PD subjects (PD-NC vs PD-MCI), SVM and RF performed poorly below the 80% threshold, with accuracy values ranging from 0–70% and 40–75% in HC and PD, respectively (S4 and S5 Tables). These observations were again not entirely surprising given that DaT-SPECT and CSF biomarkers' rates of change were not different between these cohorts (S1b Table). Additionally, DaT-SPECT is not very useful for detection of cognitive impairment in the literature, so the use of DaT-SPECT as the main neuroimaging marker may have detracted from the performance. A different neuroimaging biomarker, such as magnetic resonance imaging (MRI), may be more able to detect MCI.

Overall, our observations are consistent with some of the work done in the past. Previous studies related to parkin-sonism [17] and MCI [18] have found that higher diagnostic performance using machine learning techniques is correlated with using multiple biomarkers. For instance, one study found that a model with 96% accuracy used DaT-SPECT-derived SBRs, the t-tau/Aβ42 ratio, and University of Pennsylvania Smell Identification Test scores [17], and a different study achieved 81% accuracy using structural MRI, blood-based biomarkers, and clinical features [18]. Our finding of 100% performance using DaT, Aβ42, and p-tau for PD supports combining certain biomarkers at different stages of the disease course to improve detection of PD and similar disorders.

In general, the performance of two models (SVM and RF) was often equivalent with no clear evidence of which was the better technique, as it changed depending on the biomarkers used and the purpose of the model. Currently, diagnosis by clinicians during the early stage of PD shows approximately 84% accuracy [34]. Since both SVM and RF models for detecting PD achieved over 84%, this shows that regardless of the specific technique used, a machine learning approach can be an improvement on the current clinical methods. Because models such as SVM and RF can be applied to any set of patient data for classification, provided that the model has been suitably trained on other unrelated data, in our opin-ion, it is feasible to apply these biomarkers in a platform testing for PD and/or MCI. In any case, before these models are implemented into clinical practice, the appropriate biomarker(s) and disease timepoints would need to be tested across multiple patient groups to ensure model validity and generalizability.

There are several potential limitations to be considered. The proteomic biomarkers may only partially reflect the disease course, as the levels of the proteins in CSF and blood may change due to inflammation (e.g., NfL) [16] and protein misfolding (e.g., α-syn, Aβ42, tau) [12]. Thus, these may not be necessarily a perfect representation of the state of the disease to be measured using these modeling approaches. It is important to consider the limitation of the reduced sample size in the combined models. Many subjects did not have data for all five biofluid markers, so their data could not be used in all combined models. For instance, if a subject only had data of DaT-SPECT and p-tau, their data could not be used to train/test a model combining DaT-SPECT with NfL. In addition, because of the 70/15–20/10–15% training/validation/test split used for the combined models, the number of subjects used in the test set and to report results decreased even further. Because of this, the 100% accuracy achieved from using DaT,

Aβ42, and p-tau may be a product of overfitting. When verifying these results, robust cross-validation techniques such as nested or repeated cross-validation should be considered. Additionally, for continuing analysis, a larger sample size should be used to avoid these issues. In addition, we did not use feature selection for either the clinical or main features. Feature selection is a commonly used technique to ensure that the features used in the model contribute positively and do not detract from model performance, as some models perform poorly with high numbers of features [17]. As we did not use a large number of features, feature selection may have not been appropriate, on the other side, this approach may have improved model performance. Applying feature selection methods in future analysis should be considered. Future research should aim to improve model performance when detecting MCI in subjects with and without PD and determine whether biofluid markers can improve on the accuracy of other neuro-imaging biomarkers, such as MRI.

## Supporting information

**S1a Table. Biomarker analysis.** Abbreviations: PD; Parkinson's Disease, HC; Healthy controls, NC; Normal Cognition; MCI; Mild Cognitive Impairment; DaT; DaT-SPECT; SBR; Striatal Binding Ratio; α-syn; Alpha-synuclein; Aβ42; Beta-amyloid-42; t-tau; total-tau; p-tau; phosphorylated-tau-181; NfL; neurofilament light; Conc.; Concentration; pg; picogram; ml; millilitre, mo; month. All variables are reported with Mean (Standard Deviation). Sample sizes are reported in Table 2.
(PDF)

**S1b Table. Biomarker subgroup analysis.** Abbreviations: PD – Parkinson's Disease, HC – Healthy controls, NC – Normal Cognition; MCI – Mild Cognitive Impairment; DaT – DaT-SPECT; SBR – Striatal Binding Ratio; α-syn – Alpha-synuclein; Aβ42 – Beta-amyloid-42; t-tau – total-tau; p-tau – phosphorylated-tau-181; NfL – neurofilament light; Conc. - Concentration; pg – picogram; ml – millilitre; mo – month. All variables are reported with Mean (Standard Deviation). * Sample sizes are reported in Table 2.
(PDF)

**S2 Table. Singular Models – Metric performance for SVM and RF in HC-NC vs HC-MCI.** Abbreviations: SVM – support vector machine; RF – random forest; PD – Parkinson's Disease; MCI – Mild Cognitive Impairment; α-syn – alpha-synuclein; Aβ42 – beta-amyloid-42; t-tau – total-tau; p-tau – phosphorylated-tau; NfL – neurofilament light; ACC – accuracy, AUC – area under the curve; KPA – kappa; SNS – sensitivity; SPC – specificity.
(PDF)

**S3 Table. Singular Models – Metric performance for SVM and RF in PD-NC vs PD-MCI.** Abbreviations: SVM – support vector machine; RF – random forest; PD – Parkinson's Disease; MCI – Mild Cognitive Impairment; α-syn – alpha-synuclein; Aβ42 – beta-amyloid-42; t-tau – total-tau; p-tau – phosphorylated-tau; NfL – neurofilament light; ACC – accuracy, AUC – area under the curve; KPA – kappa; SNS – sensitivity; SPC – specificity.
(PDF)

**S4 Table. Combined Models – Metric performance for SVM and RF in HC-NC vs HC-MCI.** Abbreviations: SVM – support vector machine; RF – random forest; HC – Healthy Controls; MCI – Mild Cognitive Impairment; Aβ42 – beta-amyloid-42; p-tau – phosphorylated-tau; ACC – accuracy, AUC – area under the curve; KPA – kappa; SNS – sensitivity; SPC – specificity.
(PDF)

**S5 Table. Combined Models – Metric performance for SVM and RF in PD-NC vs PD-MCI.** Abbreviations: SVM – support vector machine; RF – random forest; MCI – Mild Cognitive Impairment; PD – Parkinson's; α-syn – alpha-synuclein;

t-tau – total-tau; p-tau – phosphorylated-tau; NfL – neurofilament light; ACC – accuracy, AUC – area under the curve; KPA – kappa; SNS – sensitivity; SPC – specificity.
(PDF)

## Acknowledgments

We would like to thank PPMI for the opportunity of using the available data.

## Author contributions

**Conceptualization:** Anthaea-Grace Patricia Dennis, Robert Chen, Antonio P. Strafella.

**Investigation:** Anthaea-Grace Patricia Dennis.

**Software:** Anthaea-Grace Patricia Dennis.

**Supervision:** Antonio P. Strafella.

**Writing – original draft:** Anthaea-Grace Patricia Dennis.

**Writing – review & editing:** Anthaea-Grace Patricia Dennis, Sarah L Martin, Robert Chen, Philip Gerretsen, Antonio P. Strafella.

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
