## [Decision Letter · Decision Letter 0]

24 Sep 2025

Dear Dr. Dennis,

Thank you for submitting your manuscript to PLOS ONE. After careful consideration, we feel that it has merit but does not fully meet PLOS ONE’s publication criteria as it currently stands. Therefore, we invite you to submit a revised version of the manuscript that addresses the points raised during the review process.

If applicable, we recommend that you deposit your laboratory protocols in protocols.io to enhance the reproducibility of your results. Protocols.io assigns your protocol its own identifier (DOI) so that it can be cited independently in the future. For instructions see: https://journals.plos.org/plosone/s/submission-guidelines#loc-laboratory-protocols . Additionally, PLOS ONE offers an option for publishing peer-reviewed Lab Protocol articles, which describe protocols hosted on protocols.io. Read more information on sharing protocols at https://plos.org/protocols?utm_medium=editorial-email&utm_source=authorletters&utm_campaign=protocols.

We look forward to receiving your revised manuscript.

Kind regards,

Gyan Prakash Modi

Academic Editor

PLOS ONE

Journal Requirements:

“A-G.P.D. is supported by the Queen Elizabeth II Graduate Scholarship in Science & Technology ( https://www.sgs.utoronto.ca/awards/queen-elizabeth-ii-graduate-scholarship-in-science-technology/), the Krembil Brain Institute (https://www.uhn.ca/Krembil), and the University of Toronto School of Graduate Studies (https://www.sgs.utoronto.ca/). A.P.S. is supported by the Canadian Institute of Health Research (CIHR) (PJT-173540) (https://cihr-irsc.gc.ca/e/193.html) and the Krembil-Rossy Chair program (https://rossy-foundation.org/;
https://www.uhn.ca/Krembil). These funders did not play any role in the study design, data collection and analysis, decision to publish, or preparation of the manuscript.”

4. Thank you for uploading your study's underlying data set. Unfortunately, the repository you have noted in your Data Availability statement does not qualify as an acceptable data repository according to PLOS's standards.

Additional Editor Comments:

Reviewer #1: The manuscript applies a machine learning approach to DaT-SPECT and CSF biomarkers for the classification of Parkinson s disease (PD) and mild cognitive impairment (MCI). The integration of imaging with fluid biomarkers is a clear strength of the work, and the use of the PPMI dataset is also commendable, as it provides high-quality and well-characterized data. With some revisions, this study has the potential to make a valuable contribution.

1. In Table 2, the descriptions of total-tau and phosphorylated-tau-181 appear to be swapped. Please verify and correct this, and confirm that the analyses themselves were not affected.

2. Although authors reported 10-fold cross-validation with a 70/30 train test split, the instances of perfect accuracy suggest possible overfitting. It would strengthen the findings to consider more rigorous strategies such as nested or repeated cross-validation, and to report confidence intervals for the performance metrics.

3. At present, only SVM and Random Forest have been applied. Including, or at least benchmarking against, an additional method such as gradient boosting or penalized regression would make the results more robust and provide better context for the reader.

4. The models also did not reliably classify MCI, so it would be best to frame the conclusions more cautiously and provide a discussion of possible biological or methodological reasons for this outcome.

5. The manuscript states that an alpha value of 0.05 was used for all tests. Please clarify if this refers to the p-value threshold, and provide more detail on how statistical tests were applied, particularly with regard to assumptions of normality and variance.

Reviewers' comments:

Reviewer's Responses to Questions

**Comments to the Author**

1. Is the manuscript technically sound, and do the data support the conclusions?

Reviewer #1: Partly

2. Has the statistical analysis been performed appropriately and rigorously?

Reviewer #1: Yes

3. Have the authors made all data underlying the findings in their manuscript fully available?

Reviewer #1: Yes

4. Is the manuscript presented in an intelligible fashion and written in standard English?

Reviewer #1: Yes

Reviewer #1: The manuscript applies a machine learning approach to DaT-SPECT and CSF biomarkers for the classification of Parkinson s disease (PD) and mild cognitive impairment (MCI). The integration of imaging with fluid biomarkers is a clear strength of the work, and the use of the PPMI dataset is also commendable, as it provides high-quality and well-characterized data. With some revisions, this study has the potential to make a valuable contribution.

1. In Table 2, the descriptions of total-tau and phosphorylated-tau-181 appear to be swapped. Please verify and correct this, and confirm that the analyses themselves were not affected.

2. Although authors reported 10-fold cross-validation with a 70/30 train test split, the instances of perfect accuracy suggest possible overfitting. It would strengthen the findings to consider more rigorous strategies such as nested or repeated cross-validation, and to report confidence intervals for the performance metrics.

3. At present, only SVM and Random Forest have been applied. Including, or at least benchmarking against, an additional method such as gradient boosting or penalized regression would make the results more robust and provide better context for the reader.

4. The models also did not reliably classify MCI, so it would be best to frame the conclusions more cautiously and provide a discussion of possible biological or methodological reasons for this outcome.

5. The manuscript states that an alpha value of 0.05 was used for all tests. Please clarify if this refers to the p-value threshold, and provide more detail on how statistical tests were applied, particularly with regard to assumptions of normality and variance.

**Do you want your identity to be public for this peer review?** For information about this choice, including consent withdrawal, please see our Privacy Policy

Reviewer #1: **Yes: ** Pankaj Pandey

---

## [Author Response · Author response to Decision Letter 1]

30 Sep 2025

Comments by Academic Editor

1) The manuscript was reformatted and the files were renamed to comply with the style requirements.

2) Code is available at https://zenodo.org/records/17195714?token=eyJhbGciOiJIUzUxMiJ9.eyJpZCI6IjQ2NzlhYzU1LWVmNjItNDc2Zi05OTQ4LWUwNjhkMjAxNjJlYyIsImRhdGEiOnt9LCJyYW5kb20iOiIwODg2ZjQ2MDQzYzFlZDMzNDA2NjgwNDQyNzFjNGU2MyJ9.11i_IRH_6RlcMpfT7JftVXTcRont4_JILk4_Er7gcHTRAfWxLDOoJJzTYj25NxeMEfRazKubunClCH7mUOiJVg. The availability of the code will change from restricted to public once the manuscript has been published.

3) The amended funding statement is in the cover letter and here:

A-G.P.D. is supported by the Queen Elizabeth II Graduate Scholarship in Science & Technology (https://www.sgs.utoronto.ca/awards/queen-elizabeth-ii-graduate-scholarship-in-science-technology/), the Krembil Brain Institute (https://www.uhn.ca/Krembil), and the University of Toronto School of Graduate Studies (https://www.sgs.utoronto.ca/). A.P.S. is supported by the Canadian Institute of Health Research (CIHR) (PJT-173540) (https://cihr-irsc.gc.ca/e/193.html) and the Krembil-Rossy Chair program (https://rossy-foundation.org/;
https://www.uhn.ca/Krembil). These funders did not play any role in the study design, data collection and analysis, decision to publish, or preparation of the manuscript. There was no additional external funding received for this study.

4) All data used in this study, as well as a data dictionary, are free and publicly available at the PPMI website (https://www.ppmi-info.org/access-data-specimens/download-data), upon an online application, the signature of the Data User Agreement and of the publications policies. For any questions on the PPMI Intellectual Property (IP) Policy or applying for an exception to the PPMI IP Policy to ppmi@michaeljfox.org. We declare that we did not have any special access privileges that others would not have when attempting to access the data from PPMI.

5) The manuscript file has been updated to include captions for supporting information files.

6) N/A

Comments by Reviewer #1

1) This has been corrected.

2) A statement on the possibility of overfitting and the use of more rigorous techniques in the future had been added. Confidence intervals were available for the 100% accuracy results, so these were added into the table reported these results.

3) We benchmarked against the accuracy of clinical diagnosis, which is approximately 84%, and used a threshold of 80% as a uniform metric (page 13). Since both SVM and RF models for detecting PD achieved over these thresholds, this shows that regardless of the specific technique used, a machine learning approach can be an improvement on the current clinical methods. Additional context was provided in the Discussion section.

4) More information on the models for MCI and possibilities for poor performance was added in the Discussion section (pages 22 – 24).

5) Additional information was provided on pages 6 – 7.

---

## [Decision Letter · Decision Letter 1]

14 Oct 2025

Using Machine Learning for Detection of Parkinson’s Disease and Mild Cognitive Impairment

PONE-D-25-36179R1

Dear Dr. Patricia Dennis,

We’re pleased to inform you that your manuscript has been judged scientifically suitable for publication and will be formally accepted for publication once it meets all outstanding technical requirements.

Kind regards,

Gyan Prakash Modi

Academic Editor

PLOS ONE

Additional Editor Comments (optional):

The authors have addressed the comments of the reviewer.

Reviewers' comments:

Reviewer's Responses to Questions

**Comments to the Author**

Reviewer #1: All comments have been addressed

2. Is the manuscript technically sound, and do the data support the conclusions?

Reviewer #1: Yes

3. Has the statistical analysis been performed appropriately and rigorously?

Reviewer #1: N/A

4. Have the authors made all data underlying the findings in their manuscript fully available?

Reviewer #1: Yes

5. Is the manuscript presented in an intelligible fashion and written in standard English?

Reviewer #1: No

Reviewer #1: (No Response)

**Do you want your identity to be public for this peer review?** For information about this choice, including consent withdrawal, please see our Privacy Policy

Reviewer #1: No

---

## [Editor Report · Acceptance letter]

PONE-D-25-36179R1

PLOS ONE

Dear Dr. Dennis,

I'm pleased to inform you that your manuscript has been deemed suitable for publication in PLOS ONE. Congratulations! Your manuscript is now being handed over to our production team.

Kind regards,

on behalf of

Dr. Gyan Prakash Modi

Academic Editor

PLOS ONE